# Food Addiction and Cognitive Functioning: What Happens in Adolescents?

**DOI:** 10.3390/nu12123633

**Published:** 2020-11-26

**Authors:** Christopher Rodrigue, Sylvain Iceta, Catherine Bégin

**Affiliations:** 1School of Psychology, Laval University, Quebec, QC G1V 0A6, Canada; Christopher.Rodrigue.1@ulaval.ca; 2Quebec Heart and Lung Institute, Laval University, Quebec, QC G1V 4G5, Canada; Sylvain.Iceta.1@ulaval.ca

**Keywords:** food addiction, adolescents, executive functioning, CANTAB, Yale Food Addiction Scale

## Abstract

This study aimed to examine cognitive factors associated to food addiction (FA) symptoms in a non-clinical sample of adolescents. A group of 25 adolescents (12–18 years; Mean age = 15.2 years) with a high level of FA symptoms (two and more) were compared to a control group without FA symptoms (*n* = 25), matched on sex and age, on four Cambridge Neuropsychological Test Automated Battery (CANTAB) neuropsychological tasks (MT: Multitasking Test; OTS: One Touch Stockings of Cambridge; SST: Stop Signal Task; RVP: Rapid Visual Information Processing). They were also compared on self-reported questionnaires assessing binge eating, depressive and anxiety symptoms, impulsivity levels, as well as executive functioning difficulties. Group comparisons did not show significant differences on neuropsychological tasks’ performances. However, effect sizes’ estimates showed small to medium effect sizes on three scores: adolescents with a high level of FA symptoms showed a higher probability of an error following an incorrect answer (OTS), a higher probability of false alarm, and a poorer target sensitivity (RVP). When referring to self-reported measurements, they reported significantly more executive functioning difficulties, more binge eating, depressive symptoms and higher impulsivity levels. Overall, results suggested that cognitive difficulties related to FA symptoms seem to manifest themselves more clearly when assessing daily activities with a self-reported questionnaire, which in turn are strongly related to overeating behaviors and psychological symptoms. Future longitudinal research is needed to examine the evolution of those variables, their relationships, and contribution in obesity onset. More precisely, the present findings highlighted the importance of affective difficulties related to this condition, as well as the need to take them into account in its assessment.

## 1. Introduction

Many authors have studied disordered eating behaviors in individuals with overweight and obesity through the lens of addictions. Indeed, they found similarities between addictive behaviors and compulsive overeating, namely behavioural and neurobiological [1,2,3,4]. These similarities led to the concept of food addiction (FA) [5]. Although there is no universal definition of FA, it is known as an excessive and abnormal intake of highly palatable foods [2,3]. Since then, the study of FA has multiplied, using the Yale Food Addiction Scale (YFAS), an instrument that consists in an adaptation of the diagnostic criteria for substance dependence, to food [2,6].

In order to deepen our understanding of the development of an addictive-like pattern of eating, it is necessary to target potential risk factors of this condition. A current hypothesis on the development of obesity suggests that deficits in executive functioning could contribute to problematic eating behaviors, attitudes towards food, and weight gain [7]. Similarly, deficits in executive functioning are considered as a central component in the development of addictive behaviors [8]. Executive function is an umbrella term including a series of cognitive processes needed to adapt to new situations, allowing to behave appropriately to the context and to produce future-oriented behaviors [9,10]. It includes central functions, like inhibition, working memory, and cognitive flexibility, which underlie many higher-order functions (e.g., reasoning, problem solving, and planning) [11,12]. Recently, a few studies have focused on the cognitive factors that could be involved in FA, mainly executive functions.

Up until now, some authors examined the cognitive factors underlying FA in adults, using neuropsychological tasks and questionnaires. In order to do so, they all compared groups of individuals endorsing higher levels of FA symptoms to individuals who endorsed fewer or none. Overall, studies conducted within adults from the general population revealed that individuals with more FA symptoms, showed faster reaction times in response to food cues among neutral pictures [13], as well as attentional biases to unhealthy food cues following a sad mood induction [14]; a poorer performance monitoring, and more difficulties to detect and process errors during the task [15]. Only one study failed to find differences between groups, on a specific task assessing inhibitory control [16]. Regarding studies in adults suffering from obesity, they essentially showed that FA symptoms were accompanied with a poorer performance on decision-making, significant deficits in sustained attention [17]; and more difficulties to detect and process errors, as well as more self-reported metacognitive difficulties [18]. Nevertheless, a very recent study failed to show specific neuropsychological impairments or difficulties in participants with FA [19]. Taken together, most of the previous studies tend to reveal that cognitive difficulties could be associated to FA symptoms. Thus, it is possible to think that these cognitive difficulties could represent risk factors and contribute to the development of FA.

Adolescence is also considered as a high-risk period to develop addictive-like behaviors, regarding the combination between less efficient emotional regulation processes and the immature impulse control, that characterize it [20]. A range of studies aimed to assess the prevalence of FA in adolescents, resulting in rates of 2 to 16% in the general population [21,22,23,24,25]; almost 17% in psychiatric inpatients and 10 to 38% in adolescents with overweight or obesity engaged in a weight-loss program [26,27,28]. These rates were similar to those observed in studies with adults [29]. Since then, a growing body of literature on the study of FA in adolescents has been observed, offering a broader understanding of this condition. For example, FA symptoms have already been found as highly correlated with more disrupted eating behaviors, as well as more impulsivity, depressive and anxiety symptoms in this population [21,22,25,26,30,31].

A recent large-scale study in adolescents aged from 12 to 18 years also showed that those with more FA symptoms also reported significantly more executive functioning difficulties on a self-reported scale (BRIEF). More precisely, they reported significantly more difficulties on both indexes, assessing behavioural regulation (inhibition, shifting, emotional control, monitoring) and metacognitive (working memory, planning/organize, organization of materials, and task completion) difficulties. It indicates that they reported more self-regulatory weakness in their everyday life, in comparison to adolescents with lower levels of FA symptoms [31]. So far, only one study has included neuropsychological tasks to assess executive functions in adolescents, according to FA symptoms. Hardee and her colleagues examined the relationships between FA symptoms and cerebral activity during an inhibitory task (Go/No-go), in a sample of adolescents [32]. Their results showed that, in participants with a higher level of FA symptoms, a hypoactivation in some brain areas was observed during the inhibitory phase of the task. They suggested that it could be associated to a poorer inhibitory control. However, no significant difference was observed when they were compared on their performances at the task. More studies are needed according to FA and cognitive functioning in adolescents, in order to identify if cognitive mechanisms are involved in addictive-like eating.

The main aim of the present study was to examine cognitive factors associated to FA symptoms in adolescents. More precisely, the objective was to assess sustained attention, as well as executive functions with neuropsychological tasks, in adolescents with a significant level of FA symptoms (two symptoms and more), and to compare them with a control group. Since very few studies have examined cognitive factors related to FA symptoms in adolescents, no formal hypothesis will be proposed.

## 2. Materials and Methods

### 2.1. Participants and Procedures

Participants were 50 adolescents (38 girls and 12 boys; Mean age = 15.20; *SD* = 1.62), selected from a larger non-clinical sample previously recruited to participate in a research study on eating behaviors in adolescents, as described in Rodrigue et al. [31]. To be included, participants were required to be aged 12–18, and have previously consent to be contacted to take part in the present study. No supplementary inclusion or exclusion criteria were added from the previous study. Within participants who accepted to be contacted subsequently for the actual study, those who reported a significant level of FA symptoms on the YFAS 2.0 (two criteria or more) were firstly contacted. This threshold represents the minimum number of criteria to endorse a FA diagnosis with a mild severity. In order to confirm the FA diagnosis, participants must additionally endorse a clinically significant impairment or distress criterion. Considering the exploratory nature of the present study and the study sample, this criterion was not taken into account in the recruitment of participants. Indeed, the objective of the study was to explore the cognitive functions of adolescents from a non-clinical sample, a population in which the early signs of FA could be observed without necessarily endorsing the diagnosis. As previously suggested in adults’ studies, the inclusion of clinically significant impairment or distress criterion may not represent a valid marker of psychological distress as some people may endorse most of the criteria and not report distress [33]. Among the initial sample, a total of 25 participants reporting a significant level of FA symptoms on the YFAS 2.0 were recruited. Throughout the manuscript, this group has been categorized as “high FA group”. Then, a comparison group of 25 adolescents who endorsed none of the FA symptoms was recruited in order to match the first group on sex and age (Control group). More precisely, for each participant from the high FA group, a control participant with the same sex and age was also recruited. Franken et al. previously used a similar procedure, in the study of the cognitive correlates of FA, in adults [15].

All participants were invited to come to our laboratory and completed, with a member of our research team trained in neuropsychological task administration, a demographic questionnaire as well as four cognitive tasks from the Cambridge Neuropsychological Test Automated Battery (CANTAB) in a quiet room without any background noises or distractions. Prior to the task completion, participants consented themselves to take part in the study and allowed us to use their previous answers to some of the questionnaires completed as part of the larger study, namely questionnaires assessing food addiction symptoms as well as binge eating, anxiety and depressive symptoms, impulsivity, and self-reported executive functioning difficulties (see Rodrigue et al.) [31]. A parental consent was required for those younger than 14 years of age. In order not to interfere in participants’ performances, they were not informed of their group belonging; a general description of the study’s objectives was given to each participant. They were also asked to self-report their height and weight, to calculate their Body Mass Index (BMI; kg/m^2^). Self-reported height and weight has previously been shown as highly correlated with measured height and weight in adolescents [34,35]. According to the World Health Organization (WHO) body mass index-for-age percentile growth charts [36], 39 participants reported a healthy weight, seven reported being overweight, four reported suffering from obesity. Both groups were evenly balanced according to the BMI categories. The Laval University Research Ethics Committee approved the study. All participants received a 20$ monetary compensation for their participation.

### 2.2. Measures

#### 2.2.1. Food Addiction

The French version of the Yale Food Addiction Scale 2.0 (YFAS 2.0) is a 35-item self-reported questionnaire used to assess food addiction symptoms over the previous year [6,37]. The YFAS 2.0 covers FA criteria based on the DSM-5 eleven diagnostic criteria for substance use disorders [38]. Items are answered on a 7-point Likert Scale ranging from 0 (Never) to 7 (Every day). To fulfill a criterion, participants must endorse at least one item related to the associated criterion. Two methods have been developed to interpret answers. First, it is possible to assess the presence/absence of the “FA diagnosis” if a participant has endorsed at least two criteria and reported functional impairment or clinical distress. The severity of the “FA diagnosis” can be categorized as “mild” (two or three criteria), “moderate” (four or five criteria), and “severe” (six or more criteria). The functional impairment/clinical distress criterion was not taken into account in the recruitment of participants. In the present study, an adapted version of the YFAS 2.0 was used to refer to age-appropriate activities, as in Rodrigue et al. (refer to Gearhardt et al. for adaptations of the YFAS for children or YFAS-C) [22,31]. To make sure that the YFAS 2.0 items were developmentally and culturally appropriate, minor adaptations have been made on the scale inspired by the YFAS-C and taking into account the specific use of the French language in the Province of Quebec. An inter-rating process was performed between two of the authors (C.R. and C.B) for each adaptation. Cronbach’s alphas for this version of the YFAS 2.0 were 0.91 for the initial study, and 0.79 for the present sample.

#### 2.2.2. Binge Eating Symptoms

The French version of the Binge Eating Scale (BES) is a 16-item self-reported questionnaire assessing symptoms related to behavioral, cognitive, and emotional manifestations of binge eating episodes [39,40]. Participants must choose, for each item, one of the four suggested sentences that most accurately described his or her situation. Each item has a designated weight on the total score, in proportion with its severity (between zero and three). The total score is obtained by summing up each item’s score, and is ranging from 0 to 46. A score of 17 or less indicates a minimal, between 18 and 26 indicated a moderate, and 27 or more indicates severe binge eating symptomatology. In the present study, Cronbach’s alpha of the BES was 0.91.

#### 2.2.3. Anxiety Symptoms

The French version of the Multidimensional Anxiety Scale For Children- Self Report (MASC) is a 39-item self-reported questionnaire assessing anxiety-related emotional, cognitive, physical and behavioural symptoms in children and adolescents from 8 to 19 years old [41,42]. Participants must answer on a 4-point Likert scale for each item, ranging from “Rarely true about me” to “Often true about me”. Items can be summed up in fours subscales: (1) Physical Symptoms, (2) Social Anxiety, (3) Harm Avoidance, and (4) Separation Anxiety. Summing subscales’ scores can also produce a global score. T-scores are created for each score and should be interpreted using the following guidelines: <40 = low; 40–54 = Average; 55–59 = High average; 60–64 = Slightly elevated; 65–69 = Elevated; ≥70 = Very elevated. In the present study, only the global score was used and showed a good internal consistency (α = 0.91).

#### 2.2.4. Depressive Symptoms

The French version of the Beck Depression Inventory (BDI) is a 21-item self-reported questionnaire assessing depressive symptoms experienced during the last two weeks [43,44]. Each symptom is rated on a 4-point Likert scale from 0 (The symptom is not associated to any suffering) to 3 (The symptom is associated to intense suffering). The total score ranges from 0 to 63; A score from 0 to 13 indicates normal to minimal depressive symptoms, a score from 14 to 19 indicates mild to moderate depressive symptoms, a score from 20 to 28 indicates moderate depressive symptoms, and a score from 29 to 63 indicates severe depressive symptoms. The BDI has previously shown good psychometric properties for large-scale screening of depressive symptoms in adolescents [45,46,47]. For the present study, Cronbach’s alpha of the questionnaire was 0.92.

#### 2.2.5. Impulsivity

The French version of the UPPS Impulsive Behaviour Scale for adolescents [48,49] is a 45-item self-reported questionnaire used to assess impulsivity in adolescents aged between 12 and 19 years old. Items are rated on a 4-point Likert scale ranging from 1 (Disagree strongly) to 4 (Agree strongly). Items are divided in four mutually exclusive dimensions of impulsivity, namely urgency, lack of premeditation, lack of perseverance and sensation seeking. A higher score indicates a higher level of impulsivity. Cronbach’s alphas for the subscales were good in the actual study, ranging from 0.82 to 0.89.

#### 2.2.6. Cognitive Functioning

Cognitive functioning was assessed using the Connect Research Suite of the Cambridge Neuropsychological Test Automated Battery (CANTAB), a computerized test battery. The CANTAB is a well-validated standardized cognitive battery that provides rapid, sensitive and objective measures of multiple cognitive domains. These tasks have been validated to understand the role of specific brain functions among a wide range of disorders. Participants completed four of the CANTAB tasks with French voiceover instructions on an iPad Air 2, namely the Multitasking Test (MT), the One Touch Stockings of Cambridge (OTS), the Rapid Visual Information Processing (RVP), and the Stop Signal Task (SST), in that order. These tasks have been developed to assess two key domains, namely sustained attention and executive functioning. For more details on the following tasks and the CANTAB battery, see cantab.com.

#### 2.2.7. Multitasking Test (MT)

MT is an eight-minute task of executive functioning that measures the participant’s ability to use multiple sources of potentially conflicting information to guide behaviour, and to ignore task-irrelevant information. During the test, an arrow can appear on both sides of the screen and point in both directions. Before each trial, a cue is displayed at the top of the screen to indicate to the participant whether they must push the right or the left button, considering the instruction to identify the side or the direction of the arrow. The task includes some sections during which the rule is consistent across trials (single task) and other sections during which the rule may randomly change from trial to trial (multitasking). The multitasking sections require a higher cognitive demand than the single task sections. Moreover, some of the task trials display congruent stimuli (e.g., the arrow on the left side is pointing on the left side of the screen) and incongruent stimuli (e.g., the arrow on the left side is pointing on the right side of the screen). The incongruent trials require a higher cognitive demand than the congruent trials. MT includes practice blocks before each assessed block, to make sure that the participants understood the instructions. The main outcomes of this task are response latency and error scores, reflecting the ability to deal with multitasking as well as the interference of incongruent task-irrelevant information on the performance at the task. This task allows calculating the cost of using interchanged rules in opposition to consistent rules, and incongruent information in opposition to congruent information. More precisely, the outcomes of interest in the present study were: the number of trials for which the outcome was incorrect; the median reaction latency of response across all correct trials; Incongruency cost (a higher incongruency cost indicates that the subject takes longer to process conflicting information); Multitasking cost (a positive score indicates that the subject responds more slowly during multitasking blocks and a higher score indicates a higher cost of managing multiple sources of information).

#### 2.2.8. One Touch Stockings of Cambridge (OTS)

OTS is a 10-min task, based upon the Tower of Hanoi test, measuring executive functioning and more precisely spatial planning and working memory. During the task, participants see two displays including three-color balls and the main instruction is to work out mentally the number of moves needed to make the lower display match the upper display. The OTS includes practice blocks to make sure that participants understood the instructions. Trials required one to six moves to complete. After an error, they are asked to rethink their solution and answer. Overall, there were 10 “easy” trials for which the upper display could be matched in one to three moves; and eight “hard” trials for which it could be matched in four to six moves. OTS outcome measures include the number of problems solved on the first choice, mean attempts required to obtain a correct solution, median latency to first choice (the median latency measured from the appearance of the stocking balls until the first choice was made, across all assessed trials where the participant’s first choice was correct), median latency to obtain a correct solution. These measures are available for all problems and/or for problems with a specified number of moves (one to six). The probabilities of an error occurring when the previous trial was responded correctly (Error given correct; the probability of an error occurring when the previous trial was responded to correctly), or incorrectly (Error given incorrect; the probability of an error occurring when the previous trial was responded to correctly), were also documented.

#### 2.2.9. Stop Signal Task (SST)

SST is a 20-min task measuring impulse control and response inhibition. It also allows measuring error monitoring after a response inhibition failure [50]. The first section of the test consists in a learning phase during which the participant must select the button associated to the direction in which the arrow points. During the second section of the test, the instruction remains the same, but when participants hear an audio tone (beep), they must hold back or inhibit their response. The SST uses a staircase design for the stop signal delay, meaning that the task allows adapting to the performance of the participant. More precisely, the stop signal delay increased by 50 ms following a successful inhibition and decreased by 50 milliseconds following a failed inhibition until the inhibition success rate stabilizes to 50%. It is suggested that the stop signal reaction time (SSRT) represents the time before which actions become ballistic and the participant is no longer able to cancel his action. The SST main outcome is the SSRT (a higher score indicates a worse stop signal reaction time). According to previous studies, performance monitoring was explored using the formula proposed by Bo, Aker, Billieux, and Landro [51]. More precisely, it allows calculating post-error slowing (PES; tendency to slow a response after a failure to stop), and post-success slowing (PSS; tendency to slow a response after a successful stop trial). PES was calculated by subtracting reaction times for “Go-after-go” trials to “Go-after-failure to stop trials”, and post success slowing (PSS) by subtracting reaction time for “Go-after-go trials” to “Go-after-successful stop trials”.

#### 2.2.10. Rapid Visual Information Processing (RVP)

The RVP is a seven-minute test assessing sustained attention and working memory. During the test, digits from 2 to 9 appear in a white box in the middle of the screen, in a pseudo-random order at the rate of 100 digits/min. The task is divided in two phases. First, participants must detect one target sequence of digits and press the button at the centre of the screen when they see the last digit of the sequence (e.g., 3–5−7); in the second phase, they must detect three target sequences of digits (e.g., 3–5−7; 2–4−6; 4–6−8), and press the button at the centre when they see the last digit of one of the sequences. RVP outcome measures include the A′, which is a metric measure representing how good the participant is at detecting a targeted sequence. RVP outcome measures also include the median response latency on trials where the subject responded correctly across all trials, and the probability of a false alarm (False alarms ÷ (False alarms + Correct rejections)).

#### 2.2.11. Behavior Rating Inventory of Executive Function–Self-Report Version (BRIEF-SR)

The French version of the BRIEF-SR is an 80-item self-reported questionnaire developed to assess adolescents’ views (11–19 years old) of their own executive functions or self-regulatory strengths and weaknesses in their everyday life [52,53]. Items are scored on a 3-point Likert scale ranging from 1 (Never) to 3 (Often), and assess the frequency of some behaviors in the last six months. Items can be regrouped in a global score (Global Executive Composite; GEC) or in two indexes (Behavioural Regulation Index (BRI); and Metacognition Index (MI)). The Behavioural Regulation Index (BRI) includes subscales for Inhibition, Shift, Emotional Control and Monitoring, and the Metacognition Index (MI) includes subscales for Working Memory, Planning/Organize, Organization of Materials and Task Completion. T-scores are created for each subscale; a score between 60 and 64 indicates mildly elevated difficulties and a score of 65 and higher indicates clinical significant difficulties. Cronbach’s alphas in the larger study were 0.83 for the BRI, 0.89 for the MI., and 0.81 for the GEC. For the present study, they were respectively of 0.80, 0.83, and 0.83.

### 2.3. Data Analysis

SPSS, version 24.0, was used for statistical analyses. Some of the variables of interest were transformed using windsorized transformation (MTT Total incorrect, MTT Incongruent cost, MTT Multitasking cost, and RVL Response latency), or using a combination of a windsorized and a logarithmic transformation (binge eating and depressive symptoms), considering their non-normal distributions [54]. Then, group comparisons were performed to compare both groups (high FA group and control group), on all variables of interest (MANOVAs and ANOVAs).

A one-way multivariate analysis of variance (MANOVA) was firstly performed to compare groups, considering the moderate correlations between the following variables: binge eating, depressive, and anxiety symptoms, as well as on impulsivity (Urgency, lack of premeditation, lack of perseverance), and self-reported executive functioning difficulties (Global Executive Composite), in order to corroborate the results from our previous study [31]. Thereafter, another one-way MANOVA was performed on all CANTAB outcomes previously described (see Measures section). More specifically, outcomes of interest were regrouped by task in the same analysis, considering their theoretical relationships under the same cognitive constructs. The latter were followed by univariate analyses of variance (ANOVAs) and descriptive analyses on all variables of interest, in order to document the severity of symptoms and difficulties in both groups. In order to quantify group comparisons’ effect sizes, partial eta-squared values were calculated and interpreted following Cohen’s guidelines (small ≥ 0.01; medium ≥ 0.06; large ≥ 0.14) [55].

## 3. Results

First, according to the YFAS 2.0 results, participants from the high FA group distributed as follows, according to the symptoms count: 10 participants reported two FA symptoms (40%), seven reported three symptoms (28%), four reported four symptoms (16%), one reported five symptoms (4%) and three participants reported six symptoms or more (12%). Within this group, four participants (16%) endorsed the FA diagnosis (i.e., two FA symptoms or more and the functional impairment or clinical distress criterion).

The initial one-way MANOVA first showed a significant difference between groups, when clustering binge eating, depressive, and anxiety symptoms, as well as impulsivity (UPPS Urgency, lack of premeditation, lack of perseverance), and self-reported executive functioning difficulties (BRIEF Global Executive Composite) (*F*(7, 39) = 5.36, *p* < 0.001, Wilks’ Λ = 0.51). Following that, One-way ANOVAs between groups showed significant differences with a medium effect size for UPPS- Lack of premeditation, and strong effect sizes for binge eating and depressive symptoms, UPPS- Urgency, UPPS- Lack of perseverance, and executive functioning difficulties. A non-significant difference with a medium effect size was found for anxiety symptoms.

A second set of one-way MANOVAs was performed to compare both groups on each CANTAB task separately, regrouping key scores previously presented. Results showed no significant differences between groups on the Multitasking Test (MTT) (*F*(4,45) = 0.45, *p* = 0.77, Wilks’ Λ = 0.96), the One Touch Stockings of Cambridge (OTS) (*F*(6,39) = 1.47, *p* = 0.21, Wilks’ Λ = 0.82), the Stop Signal Task (SST) (*F*(3,46) = 0.25, *p* = 0.86, Wilks’ Λ = 0.98), and the Rapid Visual Information Processing (RVP) (*F*(3,46) = 1.26, *p* = 0.30, Wilks’ Λ = 0.92). Interactions between groups and, sex, age (12–14 and 15–18), and BMI categories were tested independently, and were not statistically significant. However, based on independent group comparisons for each task’s outcomes, it is possible to observe three effects sizes ranging from small (OTS–Probability of error given incorrect = 0.04; RVP–Probability of false alarm = 0.03) to medium (RVP–A′ = 0.07). Descriptive statistics and ANOVAs outcomes for each dependent variable are shown in Table 1 and Table 2.

Finally, it should be noted that correlation analyses showed moderate to strong significant associations between the following self-reported variables: FA symptoms, binge eating, depressive and anxiety symptoms, UPPS Urgency scale and the BRIEF Global Executive Composite. Correlations between self-reported data and neuropsychological tasks’ outcomes were much smaller, revealing few significant associations. Among all CANTAB scores, only the Rapid Visual Information Processing (RVP) outcomes, A′ (sensitivity to the target) and the probability of false alarm, were significantly and moderately correlated with FA symptoms.

## 4. Discussion

Addictive-like eating behaviors are associated with many negative outcomes, namely weight gain, obesity and related medical conditions, a poorer quality of life, as well as more psychiatric comorbidities. Beyond this bleak picture, little is known about the risk factors that could possibly contribute to the emergence of those eating behaviors in adulthood. Thus, the present study aimed to investigate cognitive factors associated to FA symptoms in adolescents, in order to identify potential cognitive risk factors in the development of FA at this developmental stage. More specifically, the aim of this study was to compare a group of adolescents with high levels of FA symptoms (two symptoms and more), to a control group without FA symptoms, on neuropsychological tasks assessing key cognitive domains (sustained attention and executive functioning). To the best of our knowledge, our study is among the first to use neuropsychological tasks in the study of FA in adolescents.

First of all, group comparisons on self-reported data showed that participants with a high level of FA symptoms reported a significantly more severe profile on almost all self-reported assessed variables compared to adolescents from the control group. However, our results showed no statistically significant differences between groups, among key scores for all four tasks of the CANTAB neuropsychological. It means that participants from both groups showed similar performances on tasks assessing sustained attention and executive functioning. Since CANTAB cognitive tests have been highly validated as sensitive measures to examine cognitive functions related to brain networks, it is possible to think that FA symptoms in adolescents are not clearly accompanied with a specific pattern of cognitive difficulties that can be captured by neuropsychological tasks in a context of neutral stimuli. These results are supported by Hardee et al., who also noted an absence of difference on neuropsychological tasks (inhibitory control), according to FA symptoms [32]. This pattern also seems to be observed in related conditions, like binge-eating disorder (BED), which is mainly characterized by frequent overeating episodes in a discrete period of time, and a feeling of lack of control over the food intake during those episodes [38]. For example, Kittel, Schmidt, and Hilbert, only found small differences between adolescents with BED and obesity and control adolescents with obesity only, on well-validated neuropsychological tasks assessing multiple cognitive functions (sustained attention, inhibition, cognitive flexibility and decision-making) [56]. Our results as well as those of recently obtained from Hardee and Kittel’s studies may indicate that compulsive overeating behaviors are not typically associated to specific cognitive dysfunctions in this developmental stage, or that the associated cognitive difficulties cannot be captured by standard neuropsychological tasks. It is however important to note that neuroimaging findings from Hardee et al., revealed cerebral activity that could be associated with a poorer inhibitory control in participant with a higher level of FA symptoms [32]. According to this result, we cannot yet rule out the presence of cognitive difficulties or impairments in adolescents with FA symptoms.

Nevertheless, our results showed slightly poorer performances from participants within the high FA group according to effect sizes estimates, on three specific scores. Since these differences between groups were not statistically significant and effect sizes were small, the following hypotheses aim to stimulate reflection about potential cognitive factors associated to FA symptoms, according to previous findings in adults. Firstly, compared to the control group, participants from the high FA group showed a slightly higher probability of producing an error on the OTS –assessing planning and working memory – only when the previous trial was responded incorrectly. No difference between groups was observed when the previous trial was responded correctly (i.e., when the participant previously gave the expected response). It is possible to think that, after an error, it was harder for those with significant FA symptoms to adjust their behaviors in order to avoid the same error on the next trial. Previous findings, from adult studies, suggested that individuals with more FA symptoms showed more difficulties in error or performance monitoring [15,18]. Finally, both groups slightly distinguished themselves on two of the RVP scores, used to measure sustained attention. More specifically, compared to the control group, subjects from the high FA group showed a poorer target sensitivity with a medium effect size, suggesting that they were a little less efficient in detecting the sequences they were asked to; and a slightly higher probability of false alarm with a small effect size, suggesting that they tended a little bit more often to identify the target as present, when it was absent or incomplete. These outcomes could be associated with one’s difficulties to focus his attention on the ongoing task [57]. Accordingly, Steward et al. suggested deficits in sustained attention, in adults with obesity and FA [17]. Thus, FA symptoms could be associated with a poorer ability to sustain attention, as soon as in adolescence. However, these results should be replicated in order to support these hypotheses. More precisely, it would be interesting to reproduce this study in a clinical sample of adolescents endorsing the FA diagnosis, as a way to examine if the same patterns can be observed, but with larger effect sizes.

Moreover, an interesting finding is the discrepancy between group comparisons on self-reported questionnaires, and those on neuropsychological tasks. Our results showed a significant difference between groups according to self-reported executive functioning difficulties whereas no clear distinction was found on the neuropsychological tasks. Self-reported measures aim to assess the participant’s view of his own self-regulatory strengths and weaknesses in his everyday life, whereas neuropsychological tasks consist in a performance-based task designed to objectively assess cognitive functioning. Accordingly, it is possible to think that a self-reported measure of executive functioning is a more sensitive way to assess self-regulation or executive difficulties in adolescents with an addictive-like pattern of eating, while neuropsychological tasks seem not to offer a convergent source of information. Recently, Demidenko, Huntley, Martz, and Keating have argued that self-reported measures of cognitive constructs would represent a more consistent predictor of risky behaviors in adolescents (for example substance use), than cognitive tasks [58]. Moreover, they found that associations between risky behaviors and self-reported measures were stronger than with cognitive tasks outcomes. The same observation can be made from our data, as the magnitude of the associations between FA symptoms and the self-reported measure of executive functioning was clearly larger than with the neuropsychological tasks’ outcomes. Demidenko and his collaborators proposed that even though cognitive tasks predicted some risky behaviors, they seemed to require a greater power to detect a small effect, in opposition to the self-reported measures [58]. Accordingly, they reconsidered the use of neuropsychological tasks in laboratory settings as a way to infer real-world risky behaviors in adolescents. A similar pattern, in which self-reported measures of executive functioning showed clearer differences between groups than cognitive tasks, has been observed in a previous study in adults suffering from severe obesity [18].

It is possible to think that classical neuropsychological tasks do not allow capturing the affective components of executive functioning or hot executive functions. The latter are considered as major factors in the development of an addiction, involved in the weighting of pros and cons in the decision-making process [8,59]. Although the self-reported measure does not directly activate the affective components of executive functioning either, it includes items referring to emotional control in everyday life, assessing the impact of executive function difficulties on the emotional expression, as well as one’s ability to regulate or control his/her responses to emotions (e.g., “I overreact to small problems”). Even though adolescents from the high FA group did not reported clinically significant difficulties on the self-reported measure, it nevertheless allows to significantly discriminate participants according to FA symptoms. The present study highlights the relationship between self-reported executive functioning and negative affect, as depressive and anxiety symptoms, as well as with urgency. Considering the previous statements and the absence of significant difference on neuropsychological tasks, FA seems to be mainly characterized by a tendency to experiment more distress and negative affects and a tendency to act on them impulsively as well as a difficulty to regulate them satisfactorily. Besides, those could represent vulnerability factors in the development of addictive-like eating. Still, the subjective nature of self-reported measures naturally comes with personal biases that could be associated with many factors hardly measurable; it could also affect study results and come with an over or under evaluation of one’s own difficulties varying from one individual to another. Even though self-reported measures are known to provide valid findings, their subjective nature limits the associated conclusions. Thus, it would be interesting to measure the cognitive correlated of FA in adolescents in a more objective way, but in a setting with better ecological validity in order to capture the affective components of those functions in a real-life setting. We also think that a structured interview to assess eating behaviors, psychological symptoms, and impulsivity would be a great addition, to objectively assess those variables as well.

Our results should be considered in light of some limitations. First, the present sample was recruited within the general population, with a limited access to adolescents with overweight and obesity. The narrowed BMI range of the present sample has limited the inclusion of BMI categories in the analyses, as well as in the discussion of our results. In addition, the use of a subthreshold of the FA diagnosis in the recruitment of the high FA group could also represent a limitation to the present findings and conclusions. Indeed, it could have contributed to the small and statistically non-significant differences between both groups, which could have been clearer with the FA diagnosis as a threshold. Participants’ selection for the present study could also have induced a selection bias. More precisely, participants consented to be contacted for the present study, after completing a battery of questionnaires on their eating behaviors and psychological condition. It is possible that, those who presented a more severe profile, were not interested in participating in this project. A majority of the participants came from private schools, which could also have an impact on the representativeness of the sample. Moreover, since participants were received in time slots that were more convenient for them, we did not control for the moment of the testing and it could have varied across participants; it is also possible that it influenced slightly their performances, even though we asked them to pick a moment when they felt rested and awakened. Finally, the relatively small sample size, and the cross-sectional design also represented limitations in the interpretation of the outcomes. Even though the actual sample size was adequate to test our hypotheses, a larger sample size could have allowed clearer results and conclusions, especially in the examination of cognitive factors associated with FA symptoms. In order to support the present findings, future studies in this field should also include longitudinal study designs, in order to clarify the direction of the associations between psychological symptoms, cognitive functioning and FA symptoms. Beyond self-reported measures and neuropsychological tasks, neuroimaging technic would also represent an interesting addition in future studies. Finally, considering the clear relationships between FA symptoms and variables surrounding negative affect, it would also be interesting to examine neuropsychological substrates of FA, in a context involving a greater affective load.

An important strength of the present study was the use of the CANTAB, a well-validated and sensitive measure cognitive functions. The use of a computerized cognitive battery reduced the potential sources of errors across the whole process, including testing and data entry. Moreover, the use of an alternative measure of executive functioning (BRIEF-SR), offered an interesting look to the participants’ perception of their own cognitive difficulties, as compared to their objective performance on cognitive tasks. Although participants’ selection in the general population has limitations, it also represented strength of the present study. More precisely, we accessed to a group of adolescents with potentially emerging addictive-like eating behaviors accompanied with preclinical psychological symptoms. The majority of this subgroup could represent an understudied category of adolescents who seems to be in a critical period in the development of FA, during which a concerning distress is reported without being associated with important consequences. Internalized symptoms related to FA could also suggest a greater difficulty to detect and manage this condition before the development of more obvious health problems (e.g., overweight and obesity, type 2 diabetes, cardiovascular problems). Finally, the selection of a healthy control group, according to FA symptoms, also represented strength of the study design.

## 5. Conclusions

The main objective of the present study was to explore cognitive factors associated to FA symptoms in adolescents, by comparing adolescents with a significant number of symptoms to a control group, on multiple neuropsychological tasks. Even though our results did not show any significant differences or impairments on the key scores of the tasks, outcomes allowed some interesting comments and hypotheses on the potential vulnerability factors of FA in adolescents. Firstly, it highlighted the absence of clear cognitive impairment or difficulties in adolescents with a high level of FA symptoms, as assessed with neuropsychological tasks. Even though some subtle cognitive distinctions have been observed and discussed, we cannot actually conclude that cognitive difficulties are implied in the emergence of FA symptoms. Then, a prominent finding was the distinction between self-reported questionnaires and neuropsychological tasks, in the discrimination of adolescents with and without FA symptoms. Regarding this finding, we suggested that self-reported measures assessing depressive symptoms (or negative affect), impulsivity, and executive functioning difficulties were more sensitive than the computerized tasks to assess the severity of this condition. It suggests that the presence of FA symptoms in adolescents could be accompanied with a more impaired self-reported condition, including internalized symptoms, in a developmental period of major changes and distress. At this stage, the propensity to experiment negative affect and difficulties in regulating them could represent a central vulnerability factor in the development of an addictive-like pattern of eating, and should be a focal point in the future research, the clinical assessment and treatment of this condition. More studies are needed to clarify the neurocognitive mechanisms underlying FA in adolescents.

## Figures and Tables

**Table 1 nutrients-12-03633-t001:** Group comparisons–Self-reported variables (high FA group vs. control group).

	High FA Group (*n* = 25)		Control Group (*n* = 25)		Contrasts		
	M	SD	M	SD	F	P	Partial Eta-Squared
Binge eating symptoms (BES) ^a^	14 (1.10)	10.15 (0.26)	5.29 (0.70)	4.10 (0.33)			
					21.49	< 0.01	0.32
Depressive symptoms (BDI) ^a^	18.6 (1.20)	11.55 (0.31)	6.60 (0.69)	6.06 (0.45)			
					20.86	<0.01	0.32
Anxiety symptoms (MASC)	52.04	18.54	42.54	20.25	2.81	0.10	0.06
Urgency (UPPS)	30.22	6.62	23.83	6.52	11.10	<0.01	0.20
Lack of premeditation (UPPS)	24.96	5.41	21.16	5.04	6.17	<0.05	0.12
Lack of perseverance (UPPS)	22.57	4.99	17.83	4.40	11.92	<0.01	0.16
Sensation seeking (UPPS)	33.35	8.03	32.72	8.14	0.07	0.79	<0.01
Executive functioning difficulties-Global Executive Composite (GEC)	54.30	8.83	46.11	10	8.76	<0.01	0.16

*Note*: BES = Binge Eating Scale; BDI = Beck Depression Inventory; MASC = Multidimensional Anxiety Scale for Children; UPPS = Urgency, Premeditation (lack of), Perseverance (lack of), Sensation Seeking – Impulsive Behavior Scale. ^a^ Windsorized and logarithmic transformations were applied. Transformed scores are indicated in brackets.

**Table 2 nutrients-12-03633-t002:** Group comparisons–CANTAB neuropsychological tasks (high FA group vs. control group).

	High FA Group (*n* = 25)		Control Group (*n* = 25)			Contrasts	
	M	SD	M	SD	F	P	Partial Eta- Squared
**Executive functioning** MTT ^a^							
Total incorrect ^a^	7.08	6.83	6.84	6.71	0.02	0.90	<0.01
Reaction latency (ms) ^b^	550.00	83.35	565.42	83.48	0.43	0.52	<0.01
Incongruency cost ^ab^	35.86	43.92	33.84	25.07	0.04	0.84	<0.01
Multitasking cost ^ab^	199.04	96.43	181.49	107.21	0.37	0.55	<0.01
OTS							
Problems solved on first choice	10.71	2.24	10.86	1.75	0.07	0.80	<0.01
Mean choice to correct ^a^	1.48	0.36	1.45	0.25	0.12	0.73	<0.01
Latency to first choice (ms) ^b^	9172.50	4208.17	8944.32	2832.11	0.05	0.83	<0.01
Latency to correct (ms) ^b^	9827.42	4304.16	10,478.77	3773.67	0.30	0.59	<0.01
Probability of error given correct	0.31	0.17	0.32	0.13	0.11	0.75	<0.01
Probability of error given incorrect	0.26	0.22	0.19	0.18	1.69	0.20	0.04
SST							
Stop Signal Reaction Time (SSRT)	208.04	28.22	202.82	39.85	0.29	0.60	<0.01
Post-error slowing	18.14	92.27	34.02	73.81	0.45	0.51	<0.01
Post-success slowing	7.18	73.01	13.58	64.67	0.11	0.74	<0.01
**Sustained attention** RVP							
A′	0.88	0.06	0.91	0.05	3.41	0.07	0.07
Response latency (ms) ^ab^	430.78	63.17	428.62	49.46	0.02	0.89	<0.01
Probability of false alarm	0.03	0.10	0.02	0.07	1.28	0.26	0.03

*Note*: ms = milliseconds. MMT = Multitasking Test; OTS = One Touch Stockings of Cambridge; SST = Stop Signal Task; RVP = Rapid Visual Information Processing; A′ = Sensitivity to target. ^a^ Windsorized transformations were applied. ^b^ Median score.

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
