# Peer review of "Food Addiction and Cognitive Functioning: What Happens in Adolescents?"

_nutrients, 2020, doi:10.3390/nu12123633_

Round 1

Reviewer 1 Report

This is a novel study examining cognitive factors among adolescents with and without food addiction. Notably, the findings suggest few differences cognitive impairment based on measures of neuropsychological tasks, yet differences based on self-reported difficulties. The main strength of this study includes the use of a validated battery of neuropsychological assessments. I have several questions/suggestions to help strengthen the manuscript.

Methods:

  1. I was curious as to why the authors did not use the child version of the YFAS among this sample? I think that is a key limitation that should be noted in the discussion section.
  2. Were there any inclusion/exclusion criteria for the previous study not noted in this paper? If so, they should be noted since they influence the present sample.
  3. I was confusing by the two Cronbach’s alphas for the YFAS in the methods. Since the initial study findings were not reported I would suggest the authors cut that finding.

Results:

  1. It would be helpful to know what the average number of FA symptoms reported among the high food addiction group was given that a sub-threshold criterion was used. Likewise, how many of the participants endorsed distress and/or impairment and met full FA diagnostic criteria? I think this would help contextualize some of the findings.

Discussion:

  1. I think the limitation of using the sub-threshold FA diagnosis should be noted in the limitations section.
  2. I had difficulty understanding the authors interpretations in the discussion when highlighting the findings regarding errors on the OTS. The authors stated: “Firstly, compared to the control group, they showed a greater probability of producing an error on the OTS, assessing planning and working memory, only when the previous trial was responded incorrectly, with a small effect size. No difference between groups was observed when the previous trial was responded correctly.” Can the authors clarify what is meant by “when the previous trial was responded correctly?” There are also some grammatical errors throughout this paragraph which make this difficult to follow (e.g., “It is possible to think that following an error”) which should be corrected. I also think there is a typo when the authors state “DA” in this paragraph—do they mean FA? I would also caution the authors in interpreting these findings given the lack of statistical significance and suggest they soften some of the language related to findings with a small effect size. 
  3. I would recommend that the authors add “self-reported” variables assessed, in the first sentence of the second paragraph in the discussion to clarify.
  4. Finally, it might be worth discussing further that the results of this study and the findings by Hardee et al. could suggest that perhaps symptoms of food addiction are perhaps not associated with greater cognitive impairment, but perceived impairment, given the lack of differences in neuropsychological findings between groups.

Reviewer 2 Report

The manuscript is about the evaluation of executive functions in adolescents, with and without FA. The authors recruited 50 adolescents and performed a validated neuropsychological battery, after a clinical assessment of participants. I found the paper interesting and well written., focused on an important field of research, however I have some concerns that I would like to share with the authors. 1) Do you performed a power analysis before the recruitment for the evaluation of the samples size needed? 2) No differences between adolescent subgroups as regards neuropsychological tasks is common in ED literature, but the authors did not quote this aspect (that could explain the results). For example, Kittle et al. (2017) found small differences between BED adolescents and controls for EFs with different tasks. Is it possible that the developmental stage have an impact in these results or is it only linked to the tasks chosen? 3) Is it possible that self-reported instruments are biased by the evaluation of the participant in the same measure than laboratory tasks have an environment bias? 4) The authors stated in the abstract that the two groups are matched by age and sex, but I am not able to find this data in the text 5) I do not understand if this sample of adolescents were included in the Rodrigue et al. (2019) or not, because methods are the same and in the other paper the population was "not clinical". 6) Why do you not display YFAS results?

Round 2

Reviewer 2 Report

I think the authors addressed all my concerns and that the paper is really improved and clearer from a methodological perspective. I think it is suitable for publication in Nutrients

Author Response

We appreciate your time and consideration.